# VRDistill: Vote Refinement Distillation for Efficient Indoor 3D Object Detection

## ABSTRACT

Recently, indoor 3D object detection has shown impressive progress. However, these improvements have come at the cost of increased memory consumption and longer inference times, making it difficult to apply these methods in practical scenarios. To address this issue, knowledge distillation has emerged as a promising technique for model acceleration. In this paper, we propose the VRDistill framework, the first knowledge distillation framework designed for efficient indoor 3D object detection. Our VRDistill framework includes a refinement module and a soft foreground mask operation to enhance the quality of the distillation. The refinement module utilizes trainable layers to improve the quality of the teacher's votes, while the soft foreground mask operation focuses on foreground votes, further enhancing the distillation performance. Comprehensive experiments on the ScanNet and SUN-RGBD datasets demonstrate the effectiveness and generalization ability of our VRDistill framework.

## CCS CONCEPTS

• **Computing methodologies** → **Object detection**; *Shape representations*; **Learning paradigms**; • **General and reference** → Performance; • **Computer systems organization** → **Neural networks**.

## KEYWORDS

Knowledge Distillation; 3D Object Detection; Refinement

## 1 INTRODUCTION

Object detection is a fundamental task in computer vision that involves detecting and categorizing multiple semantic objects, such as vehicles, chairs, and pictures. 3D object detection, as one of the subsets of object detection, focuses on identifying and locating objects within the 3D point cloud. This task is crucial for advancing high-quality autonomous driving, auxiliary robotics, and other related fields.

As indoor 3D object detection often necessitates more precise coordinate and classification data, point cloud-based methods (e.g., VoteNet [18], H3DNet [34]) are commonly used. Recently, many attempts have been made to improve the performance of indoor 3D object detection. For example, a highly effective and representative point-based 3D object detection method is VoteNet [18], which

*ACM MM, 2024, Melbourne, Australia*
© 2024 Copyright held by the owner/author(s). Publication rights licensed to ACM.
ACM ISBN 978-x-xxxx-xxxx-x/YY/MM
https://doi.org/10.1145/nnnnnnn.nnnnnnn

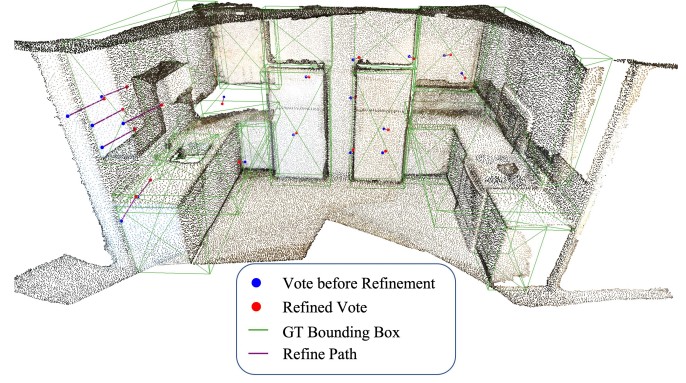

**Figure 1: Visualization of the teacher votes. Original votes from VoteNet (shown in blue) are often far away from the object centers. The votes after the refinement module shown in red are closer to the centers of the bounding boxes.**

leverages PointNet++ [20] to extract sparse point features and generate point-wise 3D proposals. However, stronger performance is often accompanied by a heavier computation burden, meaning that it is still challenging to apply existing well-performed methods to real-world applications. Meanwhile, because of the effectiveness, generality, and simplicity, knowledge distillation (KD) has become a widely-used strategy to develop efficient models in a variety of 2D tasks [8, 16]. This technique improves the performance of a lightweight and efficient student model by harvesting knowledge learned from a heavy teacher model. Despite numerous studies in 2D tasks, the investigation of KD for efficient 3D indoor object detection has largely escaped research attention.

Therefore, in this paper, we aim to design a model-agnostic framework for obtaining efficient and accurate 3D indoor object detectors based on knowledge distillation. Unlike 2D object detection, where most methods directly predict bounding boxes, indoor 3D object detection methods typically generate votes first [18] and use these votes to generate 3D bounding boxes. However, as illustrated in Fig. 1, we visualize the predicted votes from VoteNet and observe that some votes are far from the actual object centers (highlighted in blue in Fig. 1). This phenomenon is unique to 3D object detection and is caused by the property of LiDAR, which can only collect points from the objects' surface, resulting in a sparse point cloud. Consequently, the generated votes from a well-performing teacher model may not provide accurate guidance for the student model's vote generation process. In other words, the student will mimic a teacher without accurate expert knowledge, leading to ineffective distillation. Additionally, inspired by [30], the extreme foreground-background class imbalance degrades knowledge distillation in object detection. Specifically, background class losses will suppress the accurate distillation of foreground objects. Thus, it is crucial to

emphasize votes within object bounding boxes to guide distillation while simultaneously utilizing background information.

To address the aforementioned limitations, we propose a simple yet effective Vote Refinement Distillation (**VRDistill**) for indoor 3D object detection, to refine the votes from the teacher model and subsequently improve the quality of the student model. Our VRDistill framework specifically incorporates two newly proposed techniques: a refinement module and a soft foreground mask. In the refinement module, to achieve more accurate vote supervision, we initially apply a series of stacked refinement layers to the predicted votes from the teacher. Each refinement layer includes a self-attention operation to enhance contextual relations and distinctiveness for each vote cluster feature, along with a cross-attention operation for localization alignment with the initial vote clusters.

To address the second issue, we propose the soft foreground mask operation to reduce the impact of imbalance between foreground and background classes. Specifically, we assign higher mask values to the votes that are closer to the ground-truth center, representing higher confidence values for these foreground votes. Meanwhile, we assign lower mask values to the votes far away from real object centers, indicating lower confidence values for these background votes. Finally, based on the masked teacher votes and student votes, we calculate the vote consistency loss to update the student model.

Overall, our contributions are as follows:

- We propose VRDistill, the first knowledge distillation framework for efficient indoor 3D object detection, wherein the refinement module and soft foreground mask are proposed to enhance the quality of distillation.
- We propose a refinement module consisting of multiple refinement layers to improve the quality of the votes from the teacher.
- We propose a soft foreground mask operation to suppress the useless background votes and emphasize foreground votes, aiming for improved distillation performance.
- Comprehensive experiments on the ScanNet and SUN-RGBD datasets demonstrate the effectiveness and generalization ability of our VRDistill framework.

## 2 RELATED WORKS

### 2.1 3D Object Detection

Recently, several novel point-based techniques for 3D object detection [35] have emerged. These methods aim to extract features that are invariant to permutations from irregularly and sparsely distributed point cloud data. A common approach involves grouping features within a specific range, assigning a representative point to the aggregated feature, and subsequently passing it to the prediction models. Frustum PointNet [19] employs a 3D Frustum envelope generated by a 2D object detector to group points for reducing the search space, while using PointNet [20] for computing object features to predict bounding boxes. However, this dimension reduction process may result in information loss. VoteNet [18] utilizes raw point cloud data as input and adopts a simple three-phase architecture. It utilizes PointNet [20] to group point features and performs object center voting through group-wise clustering. Finally, they make predictions based on the Vote feature. Subsequent advancements to VoteNet, such as MLCVNet [22], HGNet [4], and 3DSSD

[32], have employed contextual clues, hierarchical graph neural networks, and feature-FPS sampling strategies to enhance the generation of object proposals. However, these methods heavily rely on unreliable vote clustering, which is susceptible to outliers and often overlooks inlier seed points. H3DNet [34] partially tackled this issue by introducing a hybrid set of overcomplete geometric primitives to refine the initial bounding boxes predicted by the clustered votes. Point R-CNN [25] uses deep networks to exploit the sparsity of point clouds without additional image input, but generates point-wise proposals that consume significant computation. PV-RCNN [24] uses voxel representation to complement the point-based representation in Point R-CNN for 3D object detection, resulting in improved performance. In contrast to these works, we aim to improve the performance of the lightweight 3D detection backbone based on knowledge distillation.

### 2.2 Knowledge Distillation

Knowledge distillation is a popular approach for model compression and acceleration [11]. The objective of knowledge distillation is to transfer knowledge from a powerful teacher model to a lightweight student model [10]. This technique has been widely utilized in various computer vision tasks [14, 17, 27]. Numerous knowledge distillation methods have been proposed, employing different types of representations as knowledge for enhanced performance. For instance, FitNet [23] utilizes middle-level hints from the teacher's hidden layers to guide the training of the student model. CRD [27] employs a contrastive-based objective function to transfer knowledge between deep networks. Relation-based knowledge distillation methods like CCKD [17] and RKD [14] leverage relational knowledge to improve the student model. VID [1] and PKT [15] reframe knowledge distillation as maximizing the mutual information between the teacher and student networks. The above-mentioned knowledge distillation methods for classification tasks may not apply to object detection tasks, as the spatial coordinates corresponding to the logits differ between the teacher and the student detectors. In response to this, DeFeat [8] has found that mimicking feature representations achieves better performance. [3] first proposed a method that operates on the neck feature, the classification head, and the regression head. GID [7] applies instance-wise distillation on areas where the teacher and student models perform differently. FGD [30] distills the spatial and channel features of the foreground and background of the image respectively. Inspired by MAE [9], MGD [31] shows that teacher network can improve the student's representation ability by guiding students' feature recovery. All of these methods distinguish foreground and background areas during knowledge transfer and have achieved impressive improvements. In comparison to existing knowledge distillation methods, our proposed VRDistill is specifically designed for distilling knowledge in 3D object detection. It focuses on improving the quality of votes from the teacher model and effectively transferring knowledge to the student model.

## 3 METHODOLOGY

In this section, we use VoteNet [18] as the teacher model to present our VRDistill framework. We first introduce the preliminary of

**Figure 2: Overview of VRDistill framework: Given a point cloud P, we first use the backbone to generate the teacher and student seeds $S^t$ and $S^s$. Then, we employ the vote generation module to produce teacher and student votes $V^t$ and $V^s$. In the distillation process, the refinement module refines the teacher votes, generating refined votes $R^t$, which serve as input for the foreground mask, producing masked votes $\hat{R}^t$. Simultaneously, student votes $V^s$ are aligned by the feature align module, generating aligned votes $R^s$. The vote consistency loss $L_{dis}$ is then calculated for updating the student network. Additionally, the refinement loss $L_{ref}$ is employed to facilitate vote refinement.**

VoteNet [18]. Subsequently, we will describe each part of our VRDistill framework.

## 3.1 Preliminary

VoteNet is a simple yet effective end-to-end 3D object detection method, which comprises three main components: backbone, vote generation module, and detection head. As shown in Fig. 2, given a raw point cloud $P \in \mathbb{R}^{N \times 3}$, we first use a backbone (e.g., PointNet++ [21]) to generate the seed $S$. After seed generation, we use a vote generation module to generate votes $V \in \mathbb{R}^{M \times (3+C)}$, which is implemented by multiple set abstraction module and multi-layer perceptron [18]. Here, $M$ is the number of seeds, and $C$ represents the feature dimension. Finally, we use a detection head and apply 3D Non-Maximal Suppression (3DNMS) to obtain high-quality 3D bounding boxes. More details can be found in the VoteNet [18].

## 3.2 Overview

The overview of our VRDistill framework is depicted in Fig. 2. Our goal is to use the pre-trained cumbersome teacher network to help the training of the small student network with limited capacity and achieve the best 3D object detection performance.

Given the input point cloud **P**, we first use the teacher model to generate the teacher seed $S^t$ and teacher vote $V^t$. Similarly, we also utilize the student model to generate student seeds $S^s$ and student votes $V^s$. As discussed before, the generated teacher votes $V^t$ are often far away from the real object center, leading to inaccurate distillation knowledge. Therefore, we propose to use a refinement module to refine the generated teacher votes and obtain the refined teacher votes $R^t$. At the same time, as a large number of votes lie in the background, causing a suppression of foreground votes in distillation. So we also use a soft foreground mask operation to eliminate the supervision from background votes and generate the masked teacher votes $\hat{R}^t$. Moreover, as the student model carries different feature channels from the teacher, we also use a feature align module to align the student votes and generate the aligned student votes $R^s$. Based on the masked teacher votes $\hat{R}^t$ and aligned student votes $R^s$, we calculate the vote consistency loss to update the student network as well as the refinement module. As the refinement module is randomly initialized, we also use a refinement loss to facilitate the training of the refinement module.

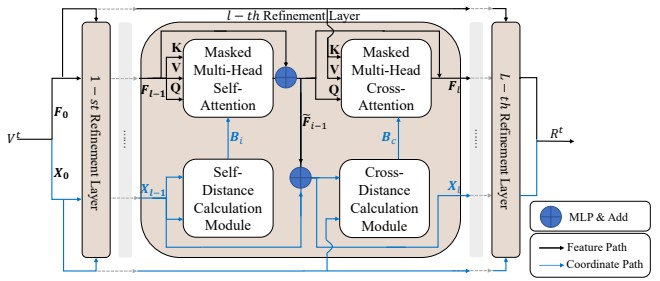

**Figure 3: The structure of the refinement module. For better presentation, we omit the addition, normalization, and feed-forward network components.**

## 3.3 Refinement Module

During the distillation process, we notice that teacher votes are often far away from the object center, inducing mismatched supervision for the student. To this end, we propose a refinement module in the distillation process to correct the inaccurate teacher votes. Specifically, the refinement module is illustrated in Fig. 3, which consists of $L$ refinement layers.

As it only takes the teacher's votes as the input, here, we omit the superscript $\cdot^t$ for better presentation. In this figure, we take the $i$-th layer as an example to introduce the structure of the refinement layer. It takes the output coordinates $\mathbf{X}_{i-1}$ and features $\mathbf{F}_{i-1}$ from the previous layer as inputs and produces output coordinates $\mathbf{X}_i$ and output features $\mathbf{F}_i$ of this layer. Specifically, we use $F_{i-1}$ as the query, key, and value to calculate a masked multi-head self-attention, where the mask is generated based on the output coordinate $X_i$. As the masked multi-head self-attention and the masked multi-head cross-attention are similar, we use masked multi-head self-attention as an example to introduce our refinement module. Note that our refinement module is trained together with the student network, so the teacher can keep the same architecture as the student network to guarantee fair comparison, and facilitate convergence during training.

**Distance calculation.** Given the output coordinates $\mathbf{X}_{i-1} \in \mathbb{R}^{M \times 3}$ from the previous layer, we first use the self-distance calculation module to calculate the intra-position information. Here, $M$ is the number of votes. We calculate the distance between any two points in $\mathbf{X}_{i-1}$ and generate the intra-position matrix $\mathbf{D}_i \in \mathbb{R}^{M \times M}$. Then, we keep the smallest $K$ values in each row of $\mathbf{D}_i$ unchanged and add extremely large negative numbers to the rest of the entries, so that they can be masked out in the attention. Mathematically, the masked intra-position information can be calculated as follows:

$$\mathbf{B}_i = \mathcal{M}(\mathcal{D}(\mathbf{X}_{i-1}, \mathbf{X}_{i-1})), \tag{1}$$

where $\mathcal{D}(\cdot, \cdot)$ is the distance calculation and $\mathcal{M}(\cdot)$ is the masking operation to add extremely small values to the rest of the entries. $\mathbf{B}_i$ denotes the masked intra-position information. The masked intra-position information $\mathbf{B}_c$ from cross-distance calculation module can be also obtained using a similar process.

**Masked multi-head self-attention.** After the intra-position information calculation, we use the output features $\mathbf{F}_{i-1}$ and masked intra-position information $\mathbf{B}_i$ to calculate the masked multi-head self-attention. We use the output features $\mathbf{F}_{i-1}$ as the query, key,

and value, and use $\mathbf{B}_i$ for masking, which can be written as follows:

$$\tilde{\mathbf{F}}_{i-1} = \tilde{\Psi}\left( \mathop{\|}_{h=1}^{H} \mathcal{S}\left( \frac{\Psi_Q^h(\mathbf{F}_{i-1})\Psi_K^h(\mathbf{F}_{i-1})^T}{\sqrt{C/H}} + \mathbf{B}_i \right) \Psi_V^h(\mathbf{F}_{i-1}) \right). \tag{2}$$

Here, $\|_{h=1}^{H}$ is the concatenation of the $H$ attentive features from different attention heads. $\mathcal{S}$ means the softmax operation along each row. $\tilde{\mathbf{F}}_{i-1}$ is the output of the masked multi-head self-attention and $H$ is the number of heads. $C$ is the number of channels in the query, the key, and the value. $\tilde{\Psi}(\cdot)$ is the linear projection operation at the end of the masked multi-head attention. $\Psi_Q^h(\cdot), \Psi_K^h(\cdot)$ and $\Psi_V^h(\cdot)$ are linear projections for the $h$-th head corresponding to the query, the key, and the value.

In this way, we generate the output of the masked multi-head self-attention $\tilde{\mathbf{F}}_{i-1}$. The operations in masked multi-head cross-attention are similar to masked multi-head self-attention except the inputs are different. After multiple refinement layers, we finally obtain refined votes $\mathbf{R}^t = [\mathbf{X}^t, \mathbf{Q}^t] \in \mathbb{R}^{M \times (3+C)}$, where $\mathbf{X}^t$ and $\mathbf{Q}^t$ are the vote coordinates and the vote features in the refined votes, respectively. The refined votes will be used for soft foreground mask operation. For better presentation, we omit addition and normalization when illustrating the refinement module. The masked-multi-head cross-attention is similar to the masked multi-head self-attention. The only difference is they use different inputs.

## 3.4 Soft Foreground Mask

Enlightened by [30] that the extreme foreground-background class imbalance induces degraded knowledge distillation on object detection, we propose to apply a foreground mask to focus on the distillation on the foreground. However, unlike 2D object detection, 3D object detection operates in a significantly sparser space, where 3D coordinates mean the actual spatial position. If we were to employ a hard mask, as many 2D object detection distillation papers do, it could potentially include low-quality information, such as noise points located at the corners of a bounding box.

To address this, based on the actual position of point cloud, we propose a soft foreground mask, which focuses more on votes that are in object bounding boxes to guide the distillation while utilizing the background information simultaneously. Mathematically, let us denote the refined votes coordinates as $\mathbf{X}^t = \{\mathbf{x}_i\}_{i=1}^{M}$, where $\mathbf{x}_i$ is the 3D coordinate for the $i$-th vote. The foreground mask for the $i$-th refined vote is as follows:

$$Mask_i = (\| \frac{\mathbf{x}_i - \mathbf{g}_j}{\mathbf{s}_j} \|_F^2 + \sigma)^{-1},$$
$$\text{where } j = \operatorname{argmin}_{j \in \{1,2,\dots,M\}} \|\mathbf{x}_i - \mathbf{g}_j\|_F^2. \tag{3}$$

$\sigma$ is a value to avoid the division of 0, which is set as 1.0 in our implementation to regulate that $Mask_i \in (0, 1]$. $\mathbf{x}_i$ is the 3D coordinate of the $i$-th refined vote, and $\mathbf{g}_j$ is the ground truth coordinate of the center for the $j$-th object. $\mathbf{s}_j \in \mathbb{R}^3$ is the length along x, y, z axes of the $j$-th ground truth box. $\| \cdot \|_F$ is the Frobenius norm.

From Eq. (3), as the distance between the vote and the nearest ground truth object center increases, the corresponding mask value decreases, the $Mask_i$ for this refined vote will decrease, indicating we assign less attention on this refined vote. By adjusting the importance of different point pairs from foreground and background

areas, as shown in Eq. (5), this foreground mask allows us to strike a balance between foreground and background information during the distillation process, enabling more effective knowledge transfer while preserving valuable background knowledge.

## 3.5 Distillation Losses

After the refinement module and soft foreground mask generation, we can use the masked votes to distill the knowledge from our teacher network. The overall loss function to train the student can be written as follows:

$$L = L_{dis} + \lambda L_{reg} + \eta L_{cls} \qquad (4)$$

where $L_{dis}$ denotes the vote consistency distillation loss between the teacher and student, which will be introduced below. $L_{reg}$ and $L_{cls}$ are the box regression and classification losses used in VoteNet, which provides accurate supervision for training the student. $\lambda$ and $\eta$ are two hyperparameters to balance different loss terms.

As 3D point clouds are unordered, one important problem in knowledge distillation is how to align the teacher and student votes. To solve this problem, we propose to use the closest vote in refined teacher vote $\mathbf{R}^t = [\mathbf{X}^t, \mathbf{Q}^t]$ as supervision, where $\mathbf{X}^t$ and $\mathbf{Q}^t$ are the coordinates and features in the refined votes. Thus, the $L_{dis}$ is calculated as follows:

$$L_{dis} = \sum_{j=1}^{M} \|\mathbf{q}_j^s - \mathbf{q}_i^t\|_F^2 \cdot Mask_i,$$

$$\text{where } i = \text{argmin}_{i \in \{1,2,...,M\}} \|\mathbf{x}_j^s - \mathbf{x}_i^t\|_F^2, \qquad (5)$$

$$\text{and } \mathbf{R}^s = Align(\mathbf{V}^s).$$

Here, $M$ is the number of votes for both student and teacher. $\mathbf{x}_i^t$ is the coordinate of $i$-th point in $\mathbf{X}^t$. $\mathbf{q}_i^t$ is the $i$-th feature in $\mathbf{Q}^t$. $Align(\cdot)$ is the projection to avoid the dimension mismatch between teacher and student votes, which is implemented by a series of MLP layers. $\mathbf{R}^s = [\mathbf{X}^s, \mathbf{Q}^s]$ is the aligned student vote, where $\mathbf{X}^s$ and $\mathbf{Q}^s$ are the coordinates and features.

As our refinement module is randomly initialized in the distillation process, we also introduce a refinement loss to provide strong supervision for better refinement performance. Suppose $\mathbf{x}_i$ is the 3D coordinates of a refined vote, the refinement loss can be written as follows:

$$L_{ref} = \frac{\sum_{i=1}^{M} \|\mathbf{x}_i - \mathbf{g}_i\| \mathbb{I}(\mathbf{s}_i \text{ on object})}{\sum_{i=1}^{M} \mathbb{I}(\mathbf{s}_i \text{ on object})} \qquad (6)$$

where $\mathbf{g}_i$ is the ground truth center of the object that the vote belongs to. $\mathbf{s}_i$ is corresponding seed that generates refined vote $\mathbf{x}_i$. $\mathbb{I}(\mathbf{s}_i \text{ on object})$ is an indicative function whether a seed $\mathbf{s}_i$ is on an object. In this way, we introduce the supervision for the refinement module to generate better-refined votes to facilitate the distillation process.

In summary, we train the student network by using Eq. (4). Meanwhile, we also use the refinement loss in Eq. (6) and Eq. (4) to update the refinement module in our VRDistill to facilitate the refinement process. In this way, we can obtain a lightweight student network with better performance.

| Setting | mAP@0.25 | mAP@0.5 |
|---|---|---|
| Teacher | 58.1 | 33.4 |
| Student (1/2) | 50.8 | 29.5 |
| Seed Disitllation | 53.0(+2.2) | 28.4(-1.1) |
| FGD [30] | 53.6(+2.8) | 30.7(+1.1) |
| MGD [31] | 53.5(+2.7) | 29.4(-0.1) |
| PGD [28] | 52.7(+1.9) | 30.8(+1.3) |
| itKD [5] | 52.7(+1.9) | 29.6(+0.1) |
| SparseKD [29] | 53.5(+2.7) | 27.1(-2.4) |
| **VRDistill (Ours)** | **58.8(+8.0)** | **36.5(+7.0)** |
| Student (1/4) | 39.9 | 20.5 |
| Seed Disitllation | 41.8(+1.9) | 21.1(+0.6) |
| FGD [30] | 43.8(+3.9) | 20.6(+0.1) |
| MGD [31] | 43.2(+3.3) | 21.9(+1.4) |
| PGD [28] | 42.0(+2.1) | 21.0(+0.5) |
| itKD [5] | 39.1(-0.8) | 16.8(-3.7) |
| SparseKD [29] | 41.3(+1.4) | 19.4(-1.1) |
| **VRDistill (Ours)** | **49.9(+10.0)** | **26.5(+6.0)** |

**Table 1: Results of different 3D indoor object detection methods on the ScanNet dataset.**

| Setting | mAP@0.25 | mAP@0.5 |
|---|---|---|
| Teacher | 57.9 | 35.7 |
| Student (1/2) | 53.2 | 26.5 |
| Seed Disitllation | 54.4(+1.2) | 28.0(+1.5) |
| FGD [30] | 53.3(+0.1) | 26.2(-0.3) |
| MGD [31] | 51.4(-1.8) | 26.1(-0.4) |
| PGD [28] | 51.6(-1.6) | 25.2(-1.3) |
| itKD [5] | 49.7(-3.5) | 23.2(-3.3) |
| SparseKD [29] | 53.9(+0.7) | 25.2(-1.3) |
| **VRDistill (Ours)** | **56.9(+3.7)** | **33.0(+6.5)** |
| Student (1/4) | 44.7 | 16.4 |
| Seed Disitllation | 47.1(+2.8) | 18.2(+1.8) |
| FGD [30] | 43.1(-1.6) | 16.1(-0.3) |
| MGD [31] | 44.0(-0.7) | 14.8(-1.6) |
| PGD [28] | 43.1(-1.6) | 16.7(+0.3) |
| itKD [5] | 36.3(-8.4) | 11.7(-4.7) |
| SparseKD [29] | 44.9(+0.2) | 15.8(-0.6) |
| **VRDistill (Ours)** | **48.7(+4.0)** | **24.8(+8.4)** |

**Table 2: Results of different 3D indoor object detection methods on the SUNRGBD dataset.**

## 4 EXPERIMENTS

In this section, we conduct comprehensive experiments to evaluate the effectiveness of our VRDistill framework.

### 4.1 Datasets

Following VoteNet, we utilize the ScanNet [6] and SUNRGBD [26] datasets to evaluate the performance of our method.

ScanNet is an RGB-D video dataset consisting of indoor scenes, specifically designed for tasks like 3D object detection, 3D instance segmentation, and semantic segmentation. It comprises 1513 scenes and 21 object categories. The dataset is divided into a training set of 1201 scenes and a validation set of 312 scenes. For evaluation purposes, we employ the standard mean Average Precision (mAP) metric, considering various Intersection over Union (IoU) thresholds, for 18 object categories. Wall, Floor, and the "Other" categories are disregarded and treated as background during the whole process. Details are shown in Tab. 6.

SUNRGBD dataset is another single-view RGB-D dataset that consists of 5285 training images and 5050 testing images for 3D scene understanding. In these 10335 scenes, approximately 50k 3D oriented bounding boxes are annotated for 37 objects. We adhere to the standard evaluation protocol and report performance on the 10 most common categories.

## 4.2 Implementation Details

We follow the experimental settings of VoteNet [18] to ensure a fair comparison. Specifically, we first pre-train the teacher model and then use our VRDistill to distill the knowledge from teacher to student. The architecture of the teacher model follows the original paper [18], while we conduct experiments by halving and quartering the number of channels in the student model.

For the feature alignment in Eq. (5), we adopt a single-layer MLP to align the feature channel between the student and the teacher. We use a raw points count of $N = 40000$ for ScanNet and 20000 for SUNRGBD. In the Vote Generation Module, we generate $M = 256$ votes. In the refinement module, we empirically set the number of refinement layers $L$ as 4 and set the number of nearest points $K$ as 5. During the distillation process, both the student and the refinement module are trained from scratch. For optimization on the ScanNet dataset, we employ the AdamW optimizer. The default weight decay is set to 0.00001, except we use 0.01 for refinement module to prevent overfitting on the student model. To better approach the global optimal point, we use a cosine learning rate schedule. We use the batch size of 8 and train student for 180 epochs. The settings on SUNRGBD are the same as those on the ScanNet dataset except that we set the initial learning rate to 0.005.

## 4.3 Experimental Results

To the best of our knowledge, there have been few studies conducted on distilling indoor 3D object detection. To show the effect of our approach, we have reimplemented several 2D object detection distillation methods including FGD [30], MGD [31], PGD [28], itKD [5] and SparseKD [29]. In this section, student (1/2) means the student network has the same network structure as the teacher but with half channels in each layer. Student (1/4) means the student network with quarter channels as the teacher in each layer.

**Results on ScanNet.** Tab. 1 shows the performance comparison of our VRDistill framework with other baseline approaches. From Tab. 1, we have the following observations:

(1) When comparing student (1/2) or student (1/4) with other distillation methods, the performance of knowledge distillation approaches surpasses training the student from scratch, which

| Refinement Module | Soft Mask | mAP@0.25 | mAP@0.5 |
|:---:|:---:|:---:|:---:|
| No | Yes | 57.6 | 34.4 |
| Yes | No | 58.5 | 35.5 |
| Yes | Yes | **58.8** | **36.5** |

**Table 3: Performance of our VRDistill when removing different components on ScanNet.**

demonstrates that it is effective to use knowledge distillation for efficient 3D indoor object detection.

(2) Our VRDistill framework outperforms other baseline methods, showing the effectiveness of our VRDistill. Specifically, our VRDistill framework outperforms the baseline method FGD by 6.1% under the quarter channel setting.

(3) It is surprising to note that our VRDistill framework even outperforms the Teacher model. We can also observe this phenomenon in [2, 33]. Besides, on ScanNet, we hypothesize that this is because the refinement module can correct the inaccurate votes in the teacher model and transfer this correction information to the student.

**Results on SUNRGBD.** To further evaluate our method, we also compare VRDistill with other distillation methods on the SUN-RGBD dataset, and the result is shown in Tab. 2. The results of our VRDistill framework indicate better performance compared with other baseline methods.

Notably, the performance improvement of students on SUN-RGBD is not as impressive as that on ScanNet. One potential explanation is as follows: compared to the ScanNet dataset, scenes in the SUNRGBD dataset are more complete. The ground truth bounding boxes in ScanNet are aligned to the axis with no angles, while in the SUNRGBD dataset, each ground truth detection box is annotated with a heading angle, making it relatively hard to achieve further improvement.

## 4.4 Ablation Study

In this section, we seek to investigate VRDistill through a series of ablation studies.

**Effect of Refinement Module.** To assess the contribution of the refinement module, we present the results of our VRDistill framework with and without the refinement module in Tab. 3. After removing the refinement module, we can observe an accuracy loss of 1.2% mAP@0.25. In the refinement module, the teacher votes are revised to be closer to the object center, thus leading to improved detection performance.

**Effect of Soft Foreground Mask.** To evaluate the contribution of the soft foreground mask, we also perform the experiments to distill the student with and without the mask, and the result is shown in Tab. 3. From the table, we can observe that the distillation with the foreground mask performs better than the distillation without the mask. This finding demonstrates the effectiveness of incorporating the foreground mask in the distillation process.

**Effect of the Number of Refinement Module layer.** In our refinement module, we set the number of transformer layers to $M = 4$. To look deep into the refinement module, we design experiments to discover the effect of the number of layers in the refinement

| Detectors | Group-Free-3D [12] | | H3DNet [34] | | 3DETR [13] | |
|---|---|---|---|---|---|---|
| Setting | mAP@0.25 | mAP@0.5 | mAP@0.25 | mAP@0.5 | mAP@0.25 | mAP@0.5 |
| Teacher | 66.7 | 48.4 | 66.3 | 48.3 | 60.6 | 42.1 |
| Student (1/2) | 52.5 | 31.5 | 62.0 | 41.0 | 58.9 | 35.4 |
| VRDistill (Ours) | 57.0(+**4.5**) | 34.7(+**3.2**) | 64.4(+**2.4**) | 46.6(+**5.6**) | 60.9(+**2.0**) | 37.5(+**2.1**) |
| Student (1/4) | 46.4 | 25.7 | 57.8 | 35.1 | 50.4 | 21.8 |
| VRDistill (Ours) | 51.8(+**5.4**) | 31.0(+**5.3**) | 58.3(+**0.5**) | 38.2(+**3.1**) | 55.8(+**5.4**) | 26.9(+**5.1**) |

**Table 4: Results of VRDistill framework when using Group-Free-3D, H3DNet, and 3DETR as backbone networks on ScanNet.**

| Detectors | Group-Free-3D [12] | | H3DNet [34] | | 3DETR [13] | |
|---|---|---|---|---|---|---|
| Setting | mAP@0.25 | mAP@0.5 | mAP@0.25 | mAP@0.5 | mAP@0.25 | mAP@0.5 |
| Teacher | 60.4 | 43.3 | 58.0 | 30.3 | 60.1 | 39.0 |
| Student (1/2) | 58.3 | 38.8 | 54.9 | 25.3 | 50.5 | 23.2 |
| VRDistill (Ours) | 60.1(+**1.8**) | 41.0(+**2.2**) | 56.2(+**1.3**) | 28.8(+**3.5**) | 52.8(+**2.3**) | 26.9(+**3.7**) |
| Student (1/4) | 56.1 | 34.1 | 50.3 | 22.6 | 43.0 | 15.5 |
| VRDistill (Ours) | 57.1(+**1.0**) | 37.1(+**3.0**) | 51.4(+**1.1**) | 25.2(+**2.6**) | 48.7(+**5.7**) | 18.4(+**2.9**) |

**Table 5: Results of VRDistill framework when using Group-Free-3D, H3DNet, and 3DETR as backbone networks on SUNRGBD.**

| Algorithm | Cabinet | Bed | Chair | Sofa | Table | Door | Window | Bookshelf | Picture | Counter | Desk | Curtain | Refrigerator | Showercurtain | Toilet | Sink | Bathtub | Garbagebin | mAP |
|---|---|---|---|---|---|---|---|---|---|---|---|---|---|---|---|---|---|---|---|
| Teacher | 35.0 | 85.3 | 87.2 | 87.2 | **61.3** | 44.6 | **35.4** | **55.1** | 3.9 | 55.7 | 65.7 | 43.9 | 43.4 | 67.1 | **98.3** | 50.1 | 90.1 | 36.3 | 58.1 |
| Student (1/2) | 27.2 | 80.3 | 84.8 | 82.8 | 54.1 | 38.2 | 28.7 | 41.0 | 1.2 | 37.0 | 60.1 | 34.5 | **48.1** | 46.8 | 93.8 | 43.2 | 89.0 | 23.4 | 50.8 |
| FGD | 28.3 | 84.6 | 85.6 | 86.2 | 54.7 | 38.7 | 33.1 | 46.3 | 3.1 | 51.1 | 62.1 | 35.1 | 37.9 | 61.2 | 93.7 | 48.2 | 82.8 | 32.4 | 53.6 |
| MGD | 24.4 | 83.9 | 84.7 | 87.7 | 55.4 | 39.8 | 29.7 | 46.6 | 3.0 | 40.5 | **67.2** | 42.4 | 42.8 | 59.3 | 91.1 | 48.2 | 87.3 | 28.9 | 53.5 |
| PGD | 24.0 | **88.1** | 84.6 | 86.5 | 56.6 | 38.5 | 29.0 | 44.4 | 1.8 | 44.6 | 65.7 | 39.2 | 43.4 | 51.3 | 92.7 | 48.0 | 80.6 | 30.6 | 52.7 |
| VRDistill (ours) | **36.7** | 86.3 | **89.8** | **88.9** | 58.5 | **50.9** | 31.7 | 46.7 | **8.4** | **60.3** | **67.2** | **51.4** | 44.5 | **67.4** | 96.3 | **55.1** | **90.2** | **37.1** | **58.8** |

**Table 6: Results of all object classes on ScanNet using 1/2 channel of VoteNet. mAP uses an IoU threshold of 0.25.**

| Number of layers ($L$) | 1 | 2 | 4 | 6 |
|---|---|---|---|---|
| mAP@0.25 | 58.1 | 57.6 | **58.8** | 57.5 |
| mAP@0.5 | 35.3 | 34.8 | **36.5** | 36.0 |

**Table 7: Results of different number of refinement layers $L$.**

| VoteNet | #Params | #FLOPs | #GPU Memory | # Time |
|---|---|---|---|---|
| Teacher | 0.95M | 5.84G | 1377.5MB | 5.03ms |
| Student (1/2) | 0.31M | 1.53G | 709.3MB | 3.47ms |
| Student (1/4) | 0.08M | 0.42G | 376.9MB | 2.94ms |

**Table 8: The complexities of different models.**

module using different values of $M$. As shown in Tab. 7, when we set $M = 1$, $M = 2$ or $M = 6$, the accuracy drops. We hypothesize that there exist under-fitting and over-fitting problems in such situations. Therefore, we set $M = 4$ by default.

**Effect of $\sigma$** In Eq. 3, the hyperparameter $\sigma$ is used to avoid the denominator to be zero. As shown in Tab. 9, we observe that when setting $\sigma = 1.0$, our VRDistill can achieve the best performance. Therefore, we choose this value as our default setting.

| $\sigma$ | 0.1 | 0.5 | 1 | 2 | 5 | 10 |
|---|---|---|---|---|---|---|
| mAP@0.25 | 58.3 | 58.0 | **58.8** | 57.3 | 56.7 | 57.7 |
| mAP@0.5 | 34.5 | 32.9 | **36.5** | 34.2 | 35.0 | 35.8 |

**Table 9: Distillation results of using different $\sigma$.**

## 4.5 Analysis

**VRDistill on Other Detectors.** To demonstrate the generalization ability of VRDistill, we evaluate our VRDistill on different backbones including Group-Free-3D [12], H3DNet [34] and 3DETR [13], and the results are shown in Tab. 4 and Tab. 5 on ScanNet and SUN-RGBD datasets. We observe consistent improvement of student network compared with vanilla training the student. It is evident that VRDistill achieves impressive results not only in VoteNet but also in other backbones, which further demonstrates the effectiveness of our proposed method.

**Performance on Different Object Classes.** To better understand our VRDistill, we report the detection performance under

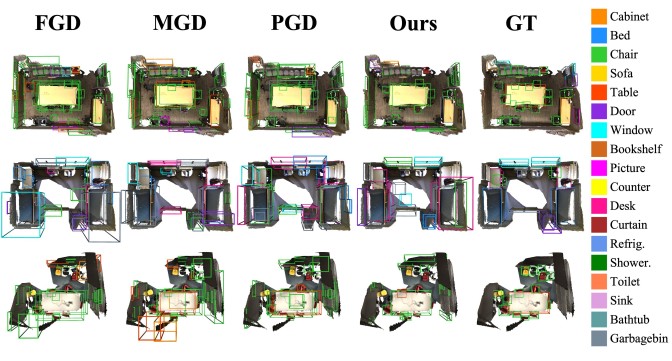

Figure 4: Visualization of results of our VRDistill framework and other baseline methods on ScanNet.

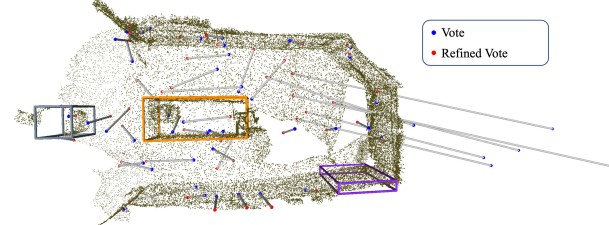

Figure 5: The refined votes are generated by the teacher model.

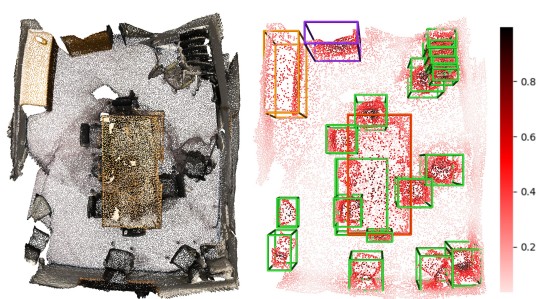

Figure 6: The heatmap of the soft mask, where the intensity of the shading corresponds to the importance of each vote. Darker shading indicates higher importance on the corresponding vote.

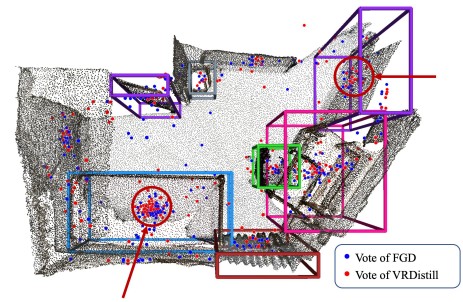

Figure 7: Vote qualities of using different methods (i.e., FGD and our proposed VRDistill).

different object classes in Tab. 6. We observe that the performance gain mainly comes from the small or thin objects (e.g., picture or showercurtain). For these objects, the votes have a higher probability of being out of the object. Therefore, our refinement module can effectively correct the inaccurate votes and provide better supervision for the student, which brings higher performance gain for these classes.

**Model Complexity.** In Table 8, we provide the number of parameters, FLOPs, and GPU memory usage, and the inference time tested using RTX3090 GPU, and we observe that the computation costs can be reduced a lot when using fewer channels. Note that we set the batch size as 1 for testing. Besides, the performance results of student models will be maintained well after using our VRDistill, which further demonstrates the effectiveness of our VRDistill.

**Visualization on detection results.** As shown in Fig. 4, we also provide the 3D object detection visualization results of different methods. When compared with the existing methods, our VRDistill can produce better results, which further demonstrates the effectiveness of our proposed method.

**Visualization on the refined votes.** As shown in Fig. 5, teacher VoteNet may generate several outliers, which may degrade the performance of distillation. After using the refinement module, we observe the votes are refined well to the foreground objects.

**Visualization on the soft foreground mask operation.** In Fig. 6, we observe that higher mask values are presented on the foreground objects, which allows us to strike a balance between foreground and background information during distillation and show the effect of soft mask operation.

**Visualization on the vote qualities between FGD and VRDistill** As shown in Fig. 7, we compare the vote quality generated from VoteNet distilled by FGD and VoteNet distilled by our VRDistill. Surprisingly, we observe that the red vote gathered more densely to the center of the bounding boxes, which means the refinement module improves the vote quality of student models in our VRDistill.

## 5 CONCLUSION

In this paper, we introduce the VRDistill framework, the first knowledge distillation framework for efficient indoor 3D object detection. Specifically, our VRDistill incorporates a refinement module and

soft foreground mask to enhance distillation quality for better performance. The refinement module leverages trainable layers to improve the quality of the teacher's votes, while the foreground mask operation focuses on foreground votes, further enhancing performance. We conducted comprehensive experiments on Scan-Net and SUN-RGBD datasets, demonstrating the effectiveness and generalization capability of our proposed VRDistill.

One of the limitation of our VRDistill is we only validate it on in-door 3D object detection methods. This is because indoor 3D object detection often necessitates more precise coordinate and classification data, which is more sensitive to the votes. We will investigate this direction in our future work.

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
