# OpenReview forum: "VRDistill: Vote Refinement Distillation for Efficient Indoor 3D Object Detection"
_acmmm.org/ACMMM/2024/Conference — MM2024 Poster_

### Official Review · Reviewer_xgzU · 2024-05-23

**Rating:** 5
**Confidence:** 2

**Summary:**

The authors introduce the VRDistill framework, a knowledge distillation framework tailored for efficient indoor 3D object detection. VRDistill incorporates a refinement module and a soft foreground mask operation to improve distillation quality. The refinement module uses trainable layers to enhance the teacher's votes, while the soft foreground mask operation targets foreground votes to further boost distillation performance. Experiments conducted on the ScanNet and SUN-RGBD datasets validate the effectiveness of the VRDistill framework.

**Strengths:**

1. The proposed method VRDistill shows superior performance on the ScanNet and SUNRGBD datasets. The authors further conduct some ablation experiments and analysis to show the effectiveness of the proposed modules and hyperparameters design.

2. The authors provide clear figures to show the details of their method and the effect of the proposed modules.

**Limitations:**

1. The authors should supplement which dataset the experiments mentioned in Table.7-9 were done on.

2. As shown in Table 7, VRDistill may not be very robust to the number of refinement layers and hyperparameters that are regarded as the most important technical contribution of this work. The author's hypothesis of underfitting and overfitting in Sec 4.4 does not fully convince me. It'd be better for the authors to 1) conduct the same experiment (set the number of layers = 1, 2, 4, 6) on another dataset to confirm the phenomenon, and 2) explore and give us more insight about the number of refinement layers.

3. The authors are encouraged to explore interesting and insightful problems: 1) is the effectiveness of the student model related to the performance of the teacher model? 2) What does the student model learn from the teacher model? (e.g., localization? orientation?...)

- In summary, I'm going to rate this work "weak accept". My minor concern lies in the robustness of the proposed method with respect to key hyperparameters such as the number of refinement layers. Also, I encourage the authors to conduct some in-depth experiments to illustrate why and how the proposed modules work, and the limitations.

**Suitability:**

2

---

### Official Review · Reviewer_T9Xf · 2024-05-24

**Rating:** 3
**Confidence:** 2

**Summary:**

This paper employs knowledge distillation techniques to process media-related information in the form of 3D point clouds, which can pave the way for novel approaches to interpreting or creating multimedia content. The proposed knowledge methods aim to advance the understanding of multimedia quality of experience through lightweight modeling, enhancing interactions and overall user satisfaction.

**Strengths:**

1. This paper proposes VRDistill, the first knowledge distillation framework for efficient indoor 3D object detection, wherein the refinement module and soft foreground mask are proposed to enhance the quality of distillation.

2.  This paper proposes a refinement module consisting of multiple refinement layers to improve the quality of the votes from the teacher.

3. This paper proposes a soft foreground mask operation to suppress the useless background votes and emphasize the foreground votes, aiming for improved distillation performance.

**Limitations:**

1. The author's motivation is that “these improvements have come at the cost of increased memory consumption and longer inference times,” however, no quantitative results with other baseline methods are provided to demonstrate the superiority of the proposed method in this aspect.

2. In addition, Table 8 only reflects a reduction in computational complexity, without a significant decrease in inference time.

3. The ablation study lacks visualization results.

4. As the author mentioned, "we only validate it on indoor 3D object detection methods," raising concerns about the method's generalization.

If the author addresses the concerns I mentioned above, I will change my perspective after rebuttal.

**Suitability:**

2

---

### Official Review · Reviewer_ozhq · 2024-05-25

**Rating:** 4
**Confidence:** 3

**Summary:**

The paper introduces VRDistill, a knowledge distillation framework designed to enhance the efficiency of indoor 3D object detection. By incorporating a refinement module and a soft foreground mask operation, the framework improves the quality of votes from a teacher model to guide the learning of a student model, thereby increasing detection performance at a lower computational cost. Comprehensive experiments on the ScanNet and SUN-RGBD datasets demonstrate the effectiveness and generalization capability of the VRDistill framework.

**Strengths:**

1. This paper exhibits a professional layout with a coherent structure, and the writing is clear and articulate, effectively conveying the research findings.
2. VRDistill is the first knowledge distillation framework designed for indoor 3D object detection, with the vote refinement module and soft foreground mask being innovative aspects.
3.The method proposed in this article has strong generalization. VRDistill not only performs well on VoteNet, but can also be generalized to other detector architectures.

**Limitations:**

1. The paper does not provide a detailed analysis of VRDistill's generalization capability across different object categories, especially for small or thin objects.

**Suitability:**

2

---

### Meta-Review · Area_Chair_v528 · 2024-06-30

**Recommendation:** Accept (Poster)
**Confidence:** 5

**Metareview:**

This paper proposes VRDistill, a framework of knowledge distillation for efficient indoor 3D object detection. As for teacher's votes, this paper uses a refinement module to enhance the quality. While for the foreground votes' quality, a soft foreground mask is proposed to suppress the useless background votes. The proposed method shows superior performance on the ScanNet and SUNRGBD datasets with ablation on the effectiveness of each proposed modules.

All three reviewers agree on the acceptance of the paper. Some concerns such as supporting evidence for the motivation and robustness are addressed in the rebuttal. AC agrees with reviewers that the paper should be accepted. Please revise the paper accordingly based on comments and incorporate new experiments provided in the rebuttal.